

# Transcriptome analysis of diploid and triploid *Populus tomentosa*

Wen Bian[1,*], Xiaozhen Liu[2,*], Zhiming Zhang[2] and Hanyao Zhang[1]

[1] Key Laboratory for Forest Resources Conservation and Utilization in the Southwest Mountains of China, Ministry of Education, Southwest Forestry University, Kunming, Yunnan, China
[2] Key Laboratory of Biodiversity Conservation in Southwest China, State Forest Administration, Southwest Forestry University, Kunming, Yunnan, China
[*] These authors contributed equally to this work.

## ABSTRACT

Triploid Chinese white poplar (*Populus tomentosa* Carr., Salicaceae) has stronger advantages in growth and better stress resistance and wood quality than diploid *P. tomentosa*. Using transcriptome sequencing technology to identify candidate transcriptome-based markers for growth vigor in young tree tissue is of great significance for the breeding of *P. tomentosa* varieties in the future. In this study, the cuttings of diploid and triploid *P. tomentosa* were used as plant materials, transcriptome sequencing was carried out, and their tissue culture materials were used for RT-qPCR verification of the expression of genes. The results showed that 12,240 differentially expressed genes in diploid and triploid *P. tomentosa* transcripts were annotated and enriched into 135 metabolic pathways. The top six pathways that enriched the most significantly different genes were plant-pathogen interaction, phenylpropanoid biosynthesis, MAPK signalling pathway-plant, ascorbate and aldarate metabolism, diterpenoid biosynthesis, and the betalain biosynthesis pathway. Ten growth-related genes were selected from pathways of plant hormone signal transduction and carbon fixation in photosynthetic organisms for RT-qPCR verification. The expression levels of *MDH* and *CYCD3* in tissue-cultured and greenhouse planted triploid *P. tomentosa* were higher than those in tissue-cultured diploid *P. tomentosa*, which was consist ent with the TMM values calculated by transcriptome.

## INTRODUCTION

*Populus tomentosa* is an important timber and ecological tree species in northern China, and it is one of the most important fast-growing native tree species in farmland shelterbelt, timber forest, and landscape tree from North China to Northwest China (*Fan et al., 2005*; *Li et al., 2020*). In terms of wood, it has good wood quality (with long fiber and white wood), high-fiber content, good paint, and cementation properties. In terms of growth characteristics, it has fast growth, long life (compared to other trees in northern China), drought and salinity resistance, strong smoke and pollution resistance, and a crown shape (*Ci et al., 2019*). Based on its excellent characteristics and the needs of ecological and

Corresponding author
Hanyao Zhang,
zhanghanyao@swfu.edu.cn,
zhanghanyao@hotmail.com

economic construction, researchers have paid more and more attention to the field of forest breeding in China (*Zong et al., 2019*).

At present, due to the problems of high genetic heterozygosity and the long breeding cycle of forest trees, the genetic improvement process of conventional cross-breeding is seriously hindered, and the productivity of varieties used in forestry production is difficult to meet the needs of industrial development for wood, biomass, paper, fuel and biomaterials (*Harfouche et al., 2012*; *Zong et al., 2019*). An in-depth study of the biological basis of wood formation and analysis of the genetic regulation mechanism of wood formations was essential to accelerate the process of improved forest varieties (*Lin et al., 2017*). The study of tree transcriptomes enables us to analyze the genetic basis of the formation of tree traits (*Hao et al., 2011*; *Zhang et al., 2019*; *Sun et al., 2019*). Because of long cycles, breeding of new varieties with expected traits cannot be rapidly achieved through traditional crossing methods alone (*Harfouche et al., 2012*). The combination of conventional breeding and modern breeding techniques represented by genomics is the main trend in the field of forest breeding in the future (*Harfouche et al., 2012*). Combining conventional breeding methods with transcriptome sequencing technology to screen tree growth-related genes is not only a breakthrough in variety selection but also a new field of exploration, and to identify candidate transcriptome-based markers for growth vigor in young tree tissue is very important (*Hao et al., 2011*; *Sun et al., 2019*).

Polyploidy is important for the evolution of plants (*Sattler, Carvalho & Clarindo, 2016*; *Liu & Sun, 2019*). RNA interference and dosage compensation in a polyploid cell often leads to epigenetic changes as it alters the gene expression levels (*Osborn et al., 2003*; *Soltis, Soltis & Tate, 2004*). Many studies have found significant differences between the expression levels of genes in diploid and polyploid plants (*Osborn et al., 2003*; *Gutierrez-Gonzalez & Garvin, 2017*; *Li et al., 2019*). Polyploid plants often exhibit commercially beneficial qualities, e.g., increased vigour, improved product quality, enlarged organs, enhanced tolerance to both biotic and abiotic stresses and increased heterozygosity and heterosis, in contrast to their diploid relatives (*Sattler, Carvalho & Clarindo, 2016*; *Liu & Sun, 2019*). Polyploidy often results in downregulated fertility because the expression of fertility-related genes is lower than that of compared to their diploid relatives (*Li et al., 2019*). Developing polyploidy in plants has been the focus of many plant breeders for some time (*Huang, Li & Cong, 1990*; *Bancroft et al., 2011*; *Rambani, Page & Udall, 2014*; *Li et al., 2019*; *Shenton et al., 2020*).

There are many advantages of triploid *P. tomentosa* plants compared with diploid ones (*Zhu et al., 1995*; *Chen et al., 2017*). Studies have shown that triploid *P. tomentosa* is superior to diploid *P. tomentosa* in both volumes per plant and papermaking (*Chen et al., 2017*; *Li & Zhang, 2000*). *P. tomentosa* has a great advantage in the construction of shelterbelt and fast-growing and high-yield forest (*Li et al., 2020*). However, the related studies on the differences in growth between diploid and triploid *P. tomentosa* are mostly focused on phenotypic studies, while there are few studies on the genes that control the differences in their growth characteristics. In this study, the transcriptome analysis was subjected to an analysis of the differences in gene expression between diploid and triploid *P. tomentosa*, which derived from genome doubling and hybridization with different

genotypes (*Zhang et al., 2008*), and RT-qPCR analysis was used for verification of the results of the transcriptome analysis. It would lay a foundation to screen growth-related genes for providing high-quality, fast-growing industrial timber and speeding up the genetic improvement of forest trees.

# MATERIALS AND METHODS

## Materials

Diploid and triploid *P. tomentosa* root sprouts were collected in Kunming World Horticultural Expo Park and on the campus of Southwest Forestry University in May 2019. These sites were adjacent to each other. The triploid poplar [(*P. tomentosa* × *P. bolleana*) × *P. tomentosa*] was produced from Beijing Forestry University (*Zhang et al., 2008*), whereas the diploid plants were native to the area. To obtain samples of the same physiological age, about one-year-old (variability in their ages about one month) well-grown root sprouts were selected and sampled at 9 a.m. The leaves were removed, and about one gram of stem segments under the fifth leaf down from the tip of the branch was sent to Wuhan Huada Gene Company for transcriptome sequencing (*Ye et al., 2020*). Among them, diploid *P. tomentosa* stem segments were used as the control group (D1, D2, D3), and triploid *P. tomentosa* stem segments were considered as the treatment group (T1, T2, T3). D1, D2, D3, T1, T2 and T3 were used to represent biological replicates.

## Sequencing data filtering

RNA was extracted using the Tiangen kit (TIANGEN Biotech Beijing Co., Ltd.), which was produced by the Wuhan Huada Gene Company. The library was generated according to the operation manual of the NEBNext® Ultra$^{TM}$ RNA Library Prep Kit for Illumina® (NEB, USA). The samples were sequenced using the Bgiseq-500 platform. The data were paired-end, and fastq formatted files were used at the beginning of the analysis. The data was filtered using Trimmomatic v0.36 software (*Bolger, Lohse & Usadel, 2014*), and Soapnuke v1.4.0 (*Chen et al., 2018*) was used for statistics. A trimming of the polyA tails was conducted, and a cut-off threshold of Q score 30 was chosen and bases with Q-score less than 30 were trimmed.

## Gene annotation

After obtaining clean reads, Bowtie2 v2.2.5 (https://sourceforge.net/projects/bowtie-bio/files/bowtie2/2.2.5/) was used to compare clean reads to *P. trichocarpa* reference genome (*Tuskan et al., 2006*) (http://popgenie.org/start?genelist=close). Then RSEM was used to calculate the expression levels of genes and transcripts (*Li & Dewey, 2011*). Seven major functional database annotations—KEGG Ortholog, KO; Gene Ontology, GO; NCBI nonredundant protein sequences, NR; NCBI nucleotide sequences, NT; A manually annotated and reviewed protein sequence database, SwissProt; Protein family, PFAM; and Clusters of Orthologous Groups of proteins, KOG—were performed on the assembled UniGenes. BLAST was used for the functional annotation of NT, Diamond was used for NR, KOG, SwissProt and KEGG, Hmmscan for PFAM and Blast2GO for GO, with the e-value of 1e −10 (*Feng et al., 2019*).

## Detection of differentially expressed genes (DEGs)

Using the transcriptome data of D1, D2, D3, T1, T2 and T3, DESeq2 software (*Anders & Huber, 2010*) was employed to screen DEGs with parameters of Fold Change $\geq 2$ and adjusted $P$-value $\leq 0.001$. The genes with a fold of more than twice in triploid *P. tomentosa* compared to diploid *P. tomentosa* and Q-value $\leq 0.001$ were screened as differentially expressed genes. QUANT (https://www.quantsoftware.com/) was used to normalise and reduce the variance and help detect DEGs. The transcripts were filtered to show the minimum count before assessing the differential expression. TMM values were calculated by using the edgeR software (*Maza, 2016*).

## Analysis of gene ontology (GO) function and KEGG function of DEGs

GO annotations were performed using nucleotide data. According to the GO annotation and KEGG annotation results, as well as the functional annotation, the differential genes were functionally classified. The detection of transcription factor families was based on these annotations. Meanwhile, the phyper function (*Evans, Hastings & Peacock, 2000*) in R software was used for enrichment analysis, the $p$-values were calculated and then corrected by False Discovery Rate (FDR) (*Burger, 2018*). Finally, the function with Q-value $\leq 0.05$ was deemed significant enrichment. The test conducted was one tailed for enrichment.

GetORF was used to detect the ORF of UniGene, while HMMER 3.0 hmmsearch was used to compare the ORF. The characteristics of the transcription factor family were then analysed by PlantTFDB (*Sanseverino et al., 2010*). The gene alignments were annotated into the plant-disease resistance gene database (PRGdb) by using DIAMOND (https://github.com/bbuchfink/diamond) v0.8.31 software (*Buchfink, Xie & Huson, 2015*).

The candidate coding region in UniGenes was identified by TransDecoder v2.0.1 software, and then the PFAM protein homologous sequence was examined by BLAST alignment SwissProt database and Hmmscan 3.0 (http://hmmer.org), thus predicting the coding region CDSs. MISA (http://pgrc.ipk-gatersleben.de/misa) was used to detect simple sequence repeats (SSR) in UniGene with the parameters set as default (*Sen et al., 2018*).

## Tissue culture of diploid and triploid *P. tomentosa*

The leaves of diploid and triploid *P. tomentosa* were inoculated on a callus induction medium (MS+1.2 mg/l 6-BA+0.6 mg/l NAA) for 15 days. After which, they were transferred to an aseptic differentiation medium (MS+1.0 mg/l 6-BA+0.4 mg/l NAA) for 28 days. Once the adventitious buds had grown by two to three cm, they were moved to a rooting medium (1/2 MS + 0.4mg/L IBA). After the adventitious roots had grown by two to three cm, the seedlings were then moved to a greenhouse.

## Real-time quantitative polymerase chain reaction (RT-qPCR) validation

To verify the reliability of transcriptome sequencing results and the expression of key genes, diploid and triploid Chinese white poplar tissue culture seedlings of the same age and growth conditions, which were derived from the same trees used for transcriptome, tissue culture plants and planted in the greenhouse, at 1 month, 4 months, 7 months, 10

**Table 1** The primers used in the RT-qPCR analysis.

| Gene symbol | Upstream and Downstream primer sequence(5→3) | Product length |
|---|---|---|
| EF1a | F:GGCAAGGAGAAGGTACACAT<br>R:CAATCACACGCTTGTCAATA | 204 |
| AUX1 | F:TGGATCTGTCATTCAACTTATTGCT<br>R:AAATACGGTAGTTATGAAAAGAGGGTAT | 143 |
| GH3 | F:GGACACCGGAAAGAAGAAGGT<br>R:CCCTGAAACATCCTAATCAAGCTAC | 211 |
| A-ARR | F:AAATCAGGGGGAGCTCTCTT<br>R:TTCTACACTTCTGTTGAGCCTGT | 100 |
| CYCD3 | F:TGCTCTGCTCTCTCTTTGTTCG<br>R:CCACTGAAAATCTCACGCCAATC | 216 |
| ABF | F:TGGAAAGTGGCAAGTGGGAA<br>R:TCAAGACACTGGCAAAGGCA | 134 |
| MDH | F:CCGGCTTCATCCACTAGACTC<br>R:GAAGGGAAGGGGTGATACCG | 153 |
| CA | F:AGAGATTATAATGGCCAGCACCAG<br>R:TGGCCCTTTTCCAGTTCCTT | 178 |
| FDP | F:ACTCCCAAACACCAAACGAGA<br>R:AGCCCACTTGGTATTGGAGC | 136 |
| SAUR | F:TGCCAAGCAAATTTTCCGCC<br>R:ACTGGAACCACAAATCGCTTC | 105 |
| ALDH | F:TGCTGGTGGACTTGAGGATT<br>R:ATCAAAGAAATGGAGAATAGGCAGA | 168 |

months and 13 months old plants, were used for RT-qPCR validation. As in transcriptome sequencing, stem segments under the fifth leaf down from the tip of a branch were selected for validation. The method of tissue culture was carried out according to the method reported by *Hu et al. (2005)*. The stem cuttings were sampled at 9 o'clock in the morning.

Ten growth-related genes were selected from the pathway of plant hormone signal transduction (Ko04075), carbon fixation in photosynthetic organisms (Ko00710), nitrogen metabolism (Ko00910) and tryptophan metabolism (Ko00380) for RT-qPCR validation. These genes were up-regulated in triploid *P. tomentosa* compared to diploid *P. tomentosa*, and taken as candidate markers for plant breeders in the future. Using the *EF1a* as an internal reference gene, and using the gene sequences of the transcriptome, the primers were designed using Primer primer 5 (*Lalitha, 2000*), as listed in Table 1. The RT-qPCR processes were described according to *Li et al. (2019)* on a Bio-Rad CFX96TM Real-time PCR Detection System (Bio-Rad, California, USA), with a final volume of 25.0 μl, containing 2.5 units of Taq DNA polymerase, 0.4 mM deoxyribonucleotides (dNTPs), 20 μl of ddH$_2$O, 1 μl of cDNA, and 500 nM of each primer. Each sample was carried out in triplicate for the RT-qPCR reactions. The data were normalised with the housekeeping gene *EF1α*, and the method was employed in line with the previous study (*Liu et al., 2018*).

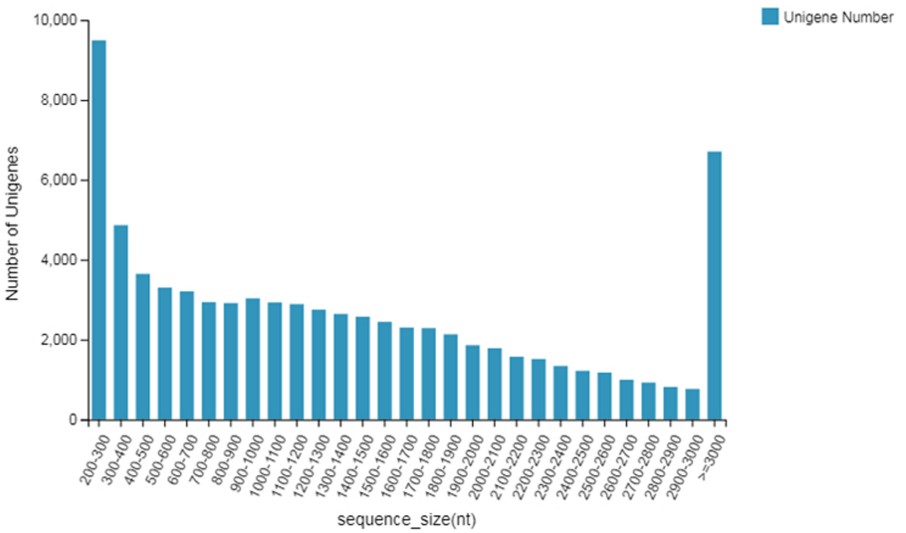

**Figure 1** **Length distribution of unigenes.**

## RESULTS AND ANALYSIS

### Unigene function annotations

The length distribution of unigenes is shown in Fig. 1. The unigenes were annotated with seven major functional database annotations (KEGG, GO, NR, NT, SwissProt, Pfam, and KOG). Finally there were 6,554 (NR: 86.43%), 71,647 (NT: 93.04%), 50,143 (SwissProt: 65.12%), 51,971 (KOG: 67.49%), 52,404 (KEGG: 68.05%), 50,210 (GO: 65.20%), and 50,881 (Pfam: 66.07%) unigenes with access to functional annotations. A total of 55,479 Coding DNA Sequences (CDSs) were detected. At the same time, 21,163 SSRs were detected distributed in 16,304 unigenes, and 3,443 unigene encoding transcription factors were predicted.

It could be seen that the annotations obtained in NT (NBCI nucleic acid sequence database) were the most, 93.04%; the overall annotation rate was 93.81%; and the lowest was 65.12% in the Swissprot database. A total of 86.43% of the annotations were obtained in the NR (NCBI protein database). According to the specific species distribution chart of the annotations, a total of five species were matched. The highest of the first three were *P. trichocarpa*, with 53.95% of the annotations, followed by *P. euphratica*, with 35.68% of the annotations, and finally, *P. tomentosa*, with 3.26% annotation.

### Transcription factor (TF) prediction

The *TF* prediction results showed that the genes belonged to a total of 55 transcription factor families, of which, with the largest number of genes, was the *MYB* gene family, with a total of 438 genes involved in the expression, followed by the *AP2-EREBP* gene family with 290 genes, and finally the *bHLH* gene family with 239 genes (see Fig. 2). Among these transcription factor families, both the *MYB* family and the *WRKY* family (including

208 genes) are involved in plant growth and development processes, which can provide relevant information for our subsequent screening of growth-related genes.

## Cluster analysis and GO classification of DEGs

The number of up-regulated genes in diploid *P. tomentosa* compared with triploid *P. tomentosa* was 15,690 and the down-regulated gene was 16,971. The scatter plot of DEGs showed that the difference of transcriptional profiles between triploid and diploid samples was obvious (see Fig. 3). The GO function was divided into three branches: molecular function, cellular component, and biological process. Figure 4 shows the functional classification based on differential gene detection.

A total of 22,375 differentially expressed genes that had GO annotations were obtained in GO classification entries by using the classification of 32,661 common differential genes. There were 13,720 DEGs in biological processes, including 6,481 up-regulated genes and 6,879 down-regulated genes; 15,963 DEGs in cell composition, including 7,687 up-regulated genes and 8,276 down-regulated genes; and 1,092 DEGs in molecular function, including 8,553 up-regulated genes and 9,539 down-regulated genes.

Among the three branches of GO function entries, the number of DEGs of binding was the largest in molecular function, the number of DEGs of the cell and cell part was the largest in the cellular component, and the number of DEGs of the cellular process was the largest in the biological process. There are 44 DEGs that belonged to lignin production-related transcripts (see Table 2), which are important for tree breeders.

A total of 38 entries were enriched in GO function ($Q \leq 0.05$) (see Table 3). Biological processes accounted for 36.8% of the total, of which catalytic activity accounted for the largest proportion of biological processes, at 42.86%; cell components accounted for 18.5% of the total, of which cell accounted for 42.85%; and molecular functions accounted for 44.7% of the total, of which catalytic activity accounted for 47.05%.

## Analysis of pathway function of DEGs

Using $Q$-value $\leq 0.05$ as the standard, 32,661 differential genes were separately analysed by pathway enrichment using the KEGG database. There were 16 significantly enriched KEGG metabolic pathways. Among them, the most frequently occupied pathways were the metabolism branch, with 13, and a total of 19,469 DEGs involved; the second was the biological system branch, with two, and a total of 1,447 DEGs involved; and finally, the environmental information process branch, with one, with 868 DEGs involved.

The results showed that there were 12,240 significantly different genes annotated in the KEGG pathways of diploid and triploid samples. The top differentially expressed genes significantly enriched six pathways were as follows: 1,115 differential gene expression in the plant-pathogen interaction (KO: ko04626) pathway; 396 differential genes in the phenylpropane biosynthesis (KO: ko00940) pathway; 868 differential genes in the mitogen-activated protein kinase (MAPK) plant signalling pathway (KO: ko04016); 170 differential genes expression in the ascorbate and aldarate metabolism pathway (KO: ko00053); 64 differential genes in the diterpenoid biosynthesis pathway (KO: ko00904); and 43 differential genes in the beets red pigment biosynthesis pathway (KO: ko00965).
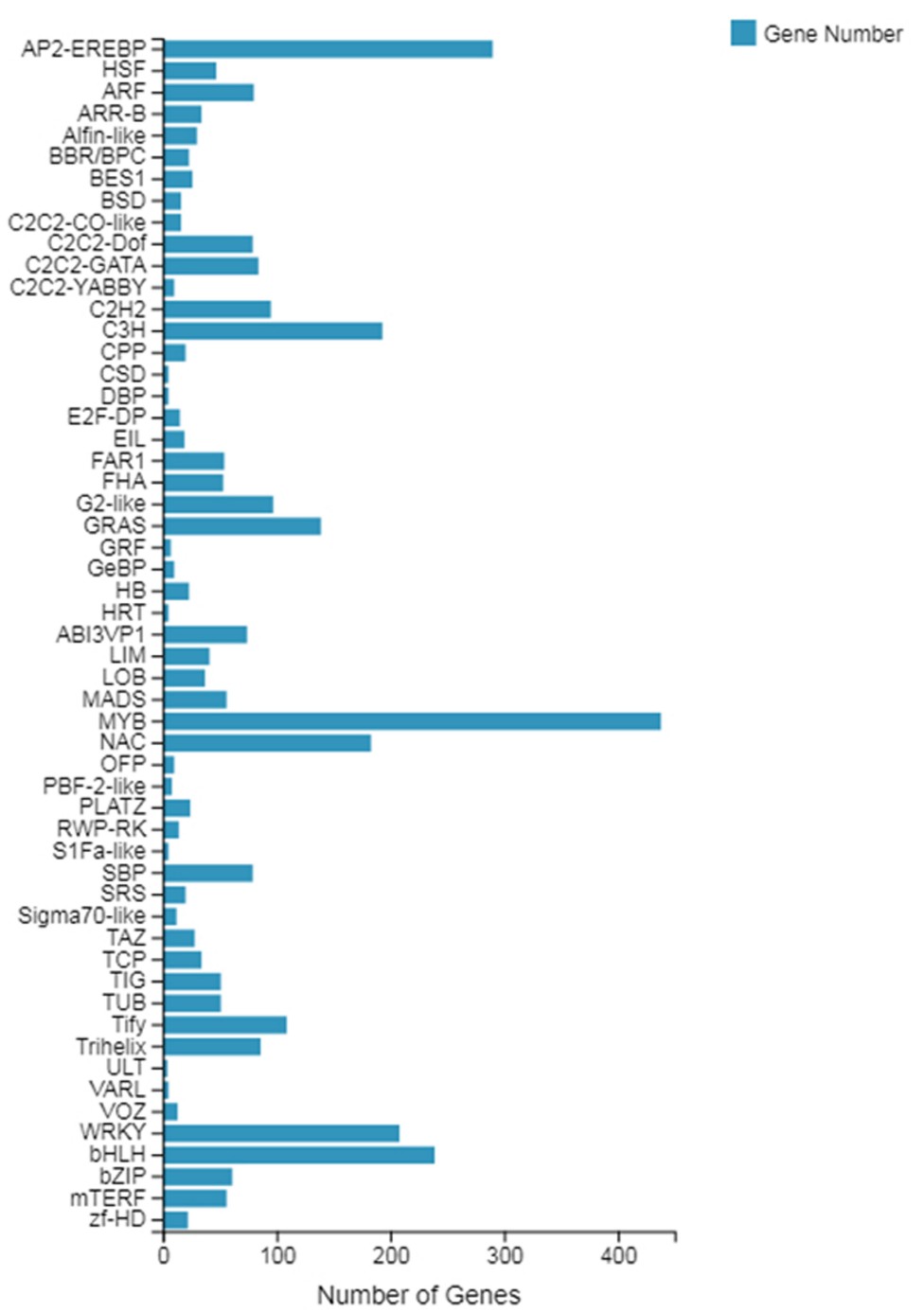

**Figure 2 Classification of the family of transcription factors.**

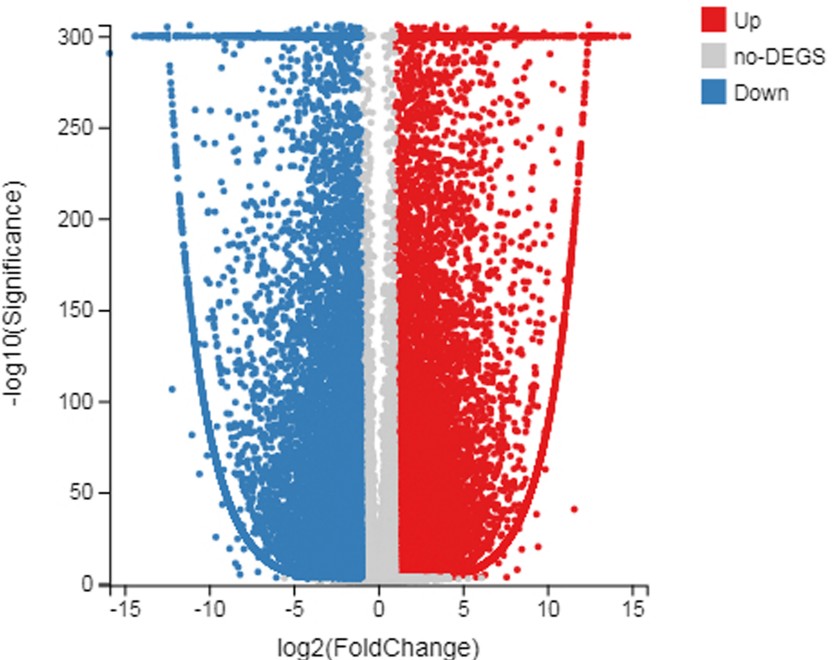

**Figure 3** **Scatter plot of log diploid and triploid expression data.** The red boxes represent upregulated transcripts after FDR; the gray boxes represent non-DEGs; the blue boxes represent downregulated transcripts.

All differentially expressed genes significantly enriched pathways were shown in Table 4. The results showed that there were 496 DEGs upregulated and 619 downregulated in the plant-pathogen interaction pathway, and that there were 864 DEGs enriched in the organismal systems category of GO annotations, 217 DEGs in environmental information processing, 25 DEGs in genetic information processing and nine DEGs in metabolism.

The differential genes in the plant-pathogen interaction (KO: ko04626) pathway could be enriched in 46 GO function entries. These GO function entries are integral components to fungus and ribosome biogenesis in the following components: the membrane, signal transduction, defence response, DNA-templated transcription, calcium ion binding, ADP binding, regulation of membrane potential, cell surface receptor signalling pathway, abscisic acid-activated signalling pathway, protein phosphorylation, peptidyl-serine phosphorylation, lipid metabolic process, intracellular signal transduction, proteolysis, protein autophosphorylation, ATP binding, plasma membrane, mRNA transcription, protein folding, 2-alkenal reductase [NAD(P)] activity, activation of protein kinase activity, kinase activity, regulation of mitotic cell cycle, calcium ion homeostasis, signal transduction by protein phosphorylation, intracellular, response to stress, membrane, extracellular region, metal ion transport, primary miRNA processing, stress-activated protein kinase signalling cascade, cellular transition metal ion homeostasis, response to temperature stimulus, calmodulin-dependent protein kinase activity, mitochondrion, endosome, protein kinase activity, developmental process involved in reproduction,

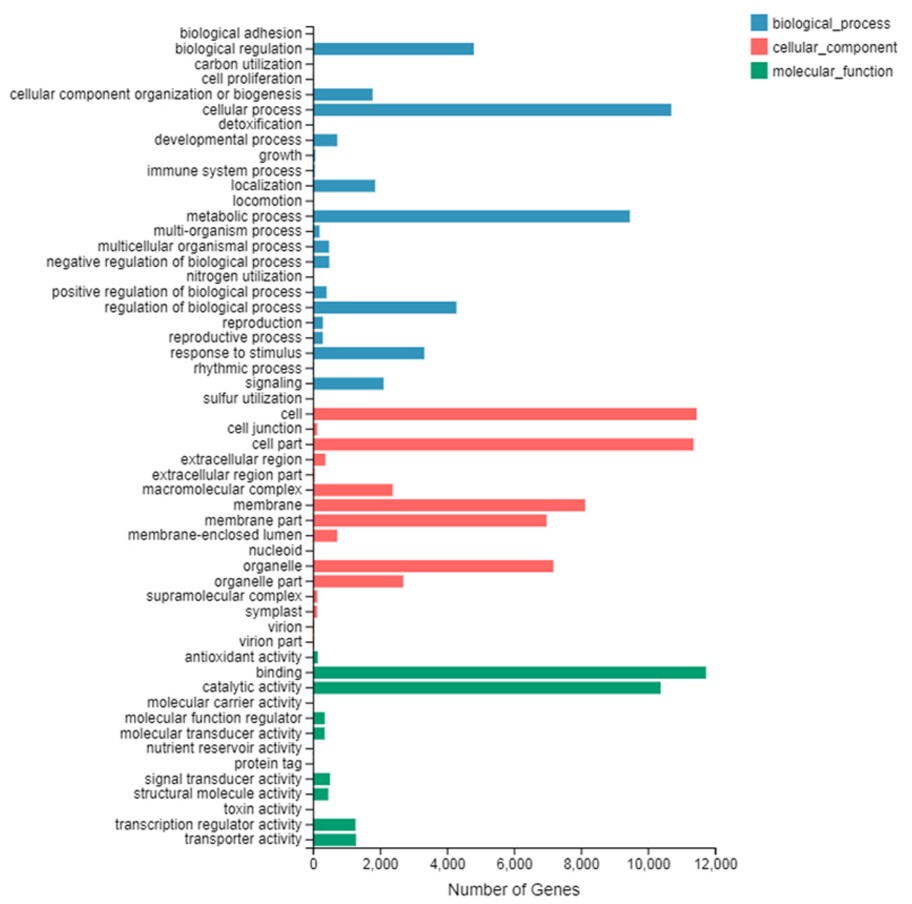

**Figure 4 GO secondary node annotation statistics of differential expression genes.** The abscissa is the number of genes and the left side of the ordinate is the GO classification.

DNA-templated regulation of transcription, plant-type hypersensitive response, defence response to bacterium, incompatible interaction, lipid catabolic process, pre-miRNA processing and defence response. Among them, the top five entries were 184 differential genes in the integral component of membrane, 174 in the signal transduction, 145 in defence response, 128 in DNA-templated transcription and 53 in calcium ion binding. In addition, there were 104 differential genes in the plant-pathogen interaction pathway that had zero expression in the triploid poplar.

In the process of analysing the growth-related pathways, it was found that the growth-related genes were up-regulated and down-regulated, so it was difficult to explain the difference between diploid and triploid samples. The NCBI database was used for gene information annotation; the details are shown in Table 5.

## RT-qPCR validation

Genes related to growth which up-regulated in the triploid *P. tomentosa* compared to diploid *P. tomentosa,* as candidate markers for plant breeders in the future, were selected for RT-qPCR validation. There was no peak in the dissolution curves of RT-qPCR

**Table 2** DEGs belonged to lignin production related transcripts.

| Gene ID | GO Term | Level 1[a] | Level 2[b] | log2(Triploid/Diploid) |
|---|---|---|---|---|
| CL3256.Contig2_All | GO:0009809//lignin biosynthetic process | Biological_process | Membrane | 1.817457211 |
| CL3256.Contig4_All | GO:0009809//lignin biosynthetic process | Cellular_component | Membrane | −1.85198633 |
| CL3256.Contig5_All | GO:0009809//lignin biosynthetic process | Molecular_function | Membrane | −1.008154399 |
| CL34.Contig3_All | GO:0046274//lignin catabolic process | Molecular_function | Extracellular region | 1.395293042 |
| CL34.Contig7_All | GO:0046274//lignin catabolic process | Biological_process | Extracellular region | 1.050358353 |
| CL34.Contig8_All | GO:0046274//lignin catabolic process | Cellular_component | Extracellular region | 4.052332527 |
| CL3608.Contig10_All | GO:0046274//lignin catabolic process | Molecular_function | Extracellular region | −3.273514187 |
| CL3608.Contig1_All | GO:0046274//lignin catabolic process | Cellular_component | Extracellular region | 4.625729696 |
| CL3608.Contig3_All | GO:0046274//lignin catabolic process | Biological_process | Extracellular region | 1.866743079 |
| CL3608.Contig6_All | GO:0046274//lignin catabolic process | Biological_process | Extracellular region | 1.65429845 |
| CL3608.Contig8_All | GO:0046274//lignin catabolic process | Biological_process | Extracellular region | 13.4946612 |
| CL3608.Contig9_All | GO:0046274//lignin catabolic process | Cellular_component | Extracellular region | −3.794382392 |
| CL4447.Contig1_All | GO:0009809//lignin biosynthetic process | Biological_process | Cellular process | 2.0065603 |
| CL4447.Contig3_All | GO:0009809//lignin biosynthetic process | Biological_process | Cellular process | 5.546749329 |
| CL4447.Contig4_All | GO:0009809//lignin biosynthetic process | Biological_process | Cellular process | 2.188480553 |
| CL4447.Contig5_All | GO:0009809//lignin biosynthetic process | Biological_process | Cellular process | 1.719058134 |
| CL4519.Contig1_All | GO:0046274//lignin catabolic process | Cellular_component | Extracellular region | −4.213801176 |
| CL4519.Contig2_All | GO:0046274//lignin catabolic process | Biological_process | Extracellular region | 2.420173712 |
| CL4519.Contig3_All | GO:0046274//lignin catabolic process | Cellular_component | Extracellular region | 2.474187548 |
| CL4519.Contig4_All | GO:0046274//lignin catabolic process | Cellular_component | Extracellular region | 1.824853683 |
| CL4519.Contig5_All | GO:0046274//lignin catabolic process | Molecular_function | Extracellular region | 1.92369436 |
| CL4519.Contig6_All | GO:0046274//lignin catabolic process | Biological_process | Extracellular region | 1.658798156 |
| CL5519.Contig1_All | GO:0046274//lignin catabolic process | Molecular_function | Extracellular region | 2.035174788 |
| CL5519.Contig2_All | GO:0046274//lignin catabolic process | Molecular_function | Extracellular region | 1.871722631 |
| CL7312.Contig1_All | GO:0046274//lignin catabolic process | Biological_process | Extracellular region | 1.766925603 |
| CL7312.Contig4_All | GO:0046274//lignin catabolic process | Cellular_component | Extracellular region | 2.018171 |
| CL7312.Contig5_All | GO:0046274//lignin catabolic process | Molecular_function | Extracellular region | 2.303556773 |
| CL7312.Contig6_All | GO:0046274//lignin catabolic process | Biological_process | Extracellular region | 1.686600197 |
| CL8534.Contig3_All | GO:0009809//lignin biosynthetic process | Biological_process | Membrane | 1.847328294 |
| CL914.Contig4_All | GO:0009808//lignin metabolic process | Cellular_component | Membrane | −2.280202421 |
| CL914.Contig5_All | GO:0009808//lignin metabolic process | Molecular_function | Membrane | −5.156601442 |
| CL9567.Contig1_All | GO:0046274//lignin catabolic process | Cellular_component | Extracellular region | −2.96038825 |
| CL9787.Contig1_All | GO:0046274//lignin catabolic process | Cellular_component | Extracellular region | 1.036570867 |
| Unigene14552_All | GO:0046274//lignin catabolic process | Biological_process | Extracellular region | 3.092109481 |
| Unigene15649_All | GO:0046274//lignin catabolic process | Cellular_component | Extracellular region | 9.382022244 |
| Unigene15651_All | GO:0046274//lignin catabolic process | Molecular_function | Extracellular region | 2.619508994 |
| Unigene17889_All | GO:0009809//lignin biosynthetic process | Molecular_function | Cellular process | −5.163595454 |
| Unigene23438_All | GO:0046274//lignin catabolic process | Cellular_component | Extracellular region | 2.998409872 |
| Unigene2393_All | GO:0046274//lignin catabolic process | Molecular_function | Extracellular region | 2.709284131 |

**Table 2** (*continued*)

| Gene ID | GO Term | Level 1[a] | Level 2[b] | log2(Triploid/Diploid) |
|---|---|---|---|---|
| Unigene24809_All | GO:0046274//lignin catabolic process | Cellular_component | Extracellular region | 4.060836332 |
| Unigene314_All | GO:0046274//lignin catabolic process | Cellular_component | Extracellular region | 2.01007043 |
| Unigene428_All | GO:0046274//lignin catabolic process | Cellular_component | Extracellular region | 1.572211841 |
| Unigene5164_All | GO:0009808//lignin metabolic process | Biological_process | Membraneactivity | −1.626207392 |
| Unigene8384_All | GO:0046274//lignin catabolic process | Biological_process | Extracellular region | 1.219895563 |

**Notes.**

[a]Large categories only includes biological process, cellular component and molecular function.

[b]Subcategories under each large category.

products of *GH3*, *CA*, and *A-ARR* gene. And the other seven genes showed a single peak curve, indicating that their amplification products did not contain primer dimers or nonspecific amplification products, and each primer PCR reaction was specific. The results of comparison of the transcriptome analysis and the RT-qPCR analysis are shown in Fig. 5A. In diploid plants, the expression levels of the *SAUR*, *FDP*, *ALDH*, *AUX1* and *ABF* in the RT-qPCR analysis were significantly higher than those in the transcriptome analysis. In triploid plants, the expression levels of the *ALDH* and *AUX1* in the RT-qPCR analysis were significantly lower than those in the transcriptome analysis, whereas the expression levels of the malate dehydrogenase (*MDH*) and *CYCD3* in the RT-qPCR analysis were significantly higher than those in the transcriptome analysis. The expression levels of *AUX1*, *CYCD3*, and *MDH* in tissue-cultured triploid poplar were higher than those of tissue-cultured diploid samples, which were consistent with the changes of TMM values calculated by transcriptome. Among them, the expression levels of the *AUX1* gene in triploid samples were significantly higher than those in diploid samples (about 6.88 times, logFC value). However, the expression of the *ABF* gene in diploid samples was higher than that of triploid samples, which was not in accordance with the TMM values. In further experiments, the expression levels of *MDH* and *CYCD3* in tissue-cultured and greenhouse planted triploid poplar were significantly higher than those of tissue-cultured and greenhouse planted diploid samples. And the expression levels of *MDH* and *CYCD3* increased with the age gradually (Figs. 5B, and 5C).

## DISCUSSION

The transcriptome analysis of the new rooting stem segments of diploid and triploid *P. tomentosa* showed that most of the significantly different genes were concentrated in plant-pathogen interaction, phenylpropane biosynthesis pathway, and MAPK signalling pathway-plant. Under the condition of comprehensive screening of the GO function enrichment and KEGG function analysis of transcriptome data, it is difficult to determine the expression of specific genes when the genes associated with plant growth appear up-regulated and down-regulated. *Huang, Li & Cong (1990)* shown there to be larger branches, leaves and fruits in the triploid variety of pear when compared to the diploid variety. And the production of tetraploid radish is 20% higher than that of ordinary diploids (*Liu, Wang & Yan, 2003*). Zhu et al. (1995) reported that allotriploidy of *P. tomentosa* had

**Table 3  Entries were enriched in GO function.**

| GO Term ID | GO Term | Level 1[a] | Level 2[b] | Rich ratio | Q-value |
|---|---|---|---|---|---|
| GO:0043531 | ADP binding | Molecular function | Binding | 0.649 | 5.07E−40 |
| GO:0007165 | Signal transduction | Biological process | Biological regulation | 0.607 | 1.54E−32 |
| GO:0006952 | Defense response | Biological process | Response to stimulus | 0.613 | 2.75E−24 |
| GO:0046914 | Transition metalion binding | Molecular function | Binding | 0.675 | 1.49E−09 |
| GO:0046916 | Cellular transition metalion homeostasis | Biological process | Biological regulation | 0.677 | 1.49E−09 |
| GO:0030001 | Metalion transport | Biological process | Localization | 0.644 | 2.10E−08 |
| GO:0005886 | Plasma membrane | Cellular component | Cell | 0.494 | 1.64E−06 |
| GO:0016021 | Integral component of membrane | Cellular component | Membrane part | 0.461 | 3.73E−06 |
| GO:0007205 | Protein kinase C-activating G-protein coupled receptor signaling pathway | Biological process | Biological regulation | 0.8 | 9.92E−05 |
| GO:0031347 | Regulation of defense response | Biological process | Biological regulation | 0.618 | 9.92E−05 |
| GO:0004143 | Diacylglycerol kinase activity | Molecular function | Catalytic activity | 0.8 | 9.99E−05 |
| GO:0006351 | Transcription, DNA-templated | Biological process | Cellular process | 0.491 | 0.001 |
| GO:0001228 | Transcriptional activator activity, RNA polymerase II transcription regulatory region sequence-specific DNA binding | Molecular function | Transcription regulator activity | 0.592 | 0.002 |
| GO:0046658 | Anchored component of plasma membrane | Cellular component | Cell | 0.565 | 0.002 |
| GO:0009611 | Response to wounding | Biological process | Response to stimulus | 0.613 | 0.002 |
| GO:0048046 | Apoplast | Cellular component | Extracellular region | 0.609 | 0.002 |
| GO:2000022 | Regulation of jasmonic acid mediated signaling pathway | Biological process | Biological regulation | 0.625 | 0.002 |
| GO:1903507 | Negative regulation of nucleic acid-templated transcription | Biological process | Biological regulation | 0.629 | 0.003 |
| GO:0000977 | RNA polymerase II regulatory region sequence-specific DNA binding | Molecular function | Binding | 0.570 | 0.003 |
| GO:0003700 | DNA binding transcription factor activity | Molecular function | Transcription regulator activity | 0.497 | 0.003 |
| GO:0003839 | Gamma-glutamylcyclotransferase activity | Molecular function | Catalytic activity | 0.941 | 0.004 |
| GO:0006979 | Response to oxidative stress | Biological process | Response to stimulus | 0.596 | 0.006 |
| GO:0005576 | Extracellular region | Cellular component | Extracellular region | 0.541 | 0.006 |
| GO:0016298 | Lipase activity | Molecular function | Catalytic activity | 0.698 | 0.006 |

**Table 3** (*continued*)

| GO Term ID | GO Term | Level 1[a] | Level 2[b] | Rich ratio | *Q*-value |
|---|---|---|---|---|---|
| GO:0016020 | Membrane | Cellular component | Membrane | 0.501 | 0.007 |
| GO:0006629 | Lipid metabolic process | Biological process | Metabolic process | 0.550 | 0.010 |
| GO:0016747 | Transferase activity, transferring acyl groups other than amino-acyl groups | Molecular function | Catalytic activity | 0.578 | 0.012 |
| GO:0001085 | RNA polymerase II transcription factor binding | Molecular function | Binding | 0.635 | 0.018 |
| GO:0031408 | Oxylipin biosynthetic process | Biological process | Cellular process | 0.627 | 0.018 |
| GO:0004190 | Aspartic-type endopeptidase activity | Molecular function | Catalytic activity | 0.559 | 0.021 |
| GO:0020037 | Heme binding | Molecular function | Binding | 0.521 | 0.0211 |
| GO:0003714 | Transcription corepressor activity | Molecular function | Transcription regulator activity | 0.589 | 0.023 |
| GO:0004612 | Phosphoenolpyruvate carboxykinase (ATP) activity | Molecular function | Catalytic activity | 0.786 | 0.027 |
| GO:0003951 | NAD+ kinase activity | Molecular function | Catalytic activity | 0.638 | 0.0418 |
| GO:0004601 | Peroxidase activity | Molecular function | Antioxidant activity | 0.607 | 0.0418 |
| GO:0006814 | Sodium ion transport | Biological process | Localization | 0.778 | 0.0418 |
| GO:0005618 | Cell wall | Cellular component | Cell | 0.536 | 0.042 |
| GO:0009041 | Uridylate kinase activity | Molecular function | Catalytic activity | 0.842 | 0.042 |

**Notes.**
[a]Large categories only includes biological process, cellular component and molecular function.
[b]Subcategories under each large category.

greater values than diploidy under the same growth conditions in tree height, diameter at breast height, and single plant volume at the age of eight years.

It is interesting that although the transcriptome of the diploidy and triploidy of *P. tomentosa* was analysed here, according to the annotations of the specific species distribution chart, only 3.26% was annotated by *P. tomentosa*. This may partly be due to the fact that the triploid materials used in this study were obtained by the hybridised *P. bolleana*, and some of the mRNAs did not belong in *P. tomentosa*. Of course, it may also partly be due to the fact that the plant stems were taken from annual plants in this study, which is different to previous studies on *P. tomentosa*. Previous studies on the transcriptome analysis of *P. tomentosa* were mostly based on aseptic seedlings, root one-month-old stem and leaf sample materials (*An et al., 2011*; *Wang, Wu & Bo, 2018*), and the expression of mRNA was organ-specific (*Ohtsuki et al., 2005*).

The phytohormone signal transduction pathway controls plant cell division, cell elongation, cell enlargement, and stem elongation, which is closely related to plant growth and development (*Guo et al., 2020*). Compared with previous data of transcriptome analysis on phytohormone signal transduction, we selected *AUX1*, *GH3*, *A-ARR*, *CYCD3*, *ABF* and five other genes for the RT-qPCR validation. Huge differences were found

**Table 4  Differentially expressed genes significantly enriched KEGG pathways.**

| Pathway ID | Pathway name | Level 1[a] | Level 2[b] | Rich ratio | Q value |
|---|---|---|---|---|---|
| ko04626 | Plant-pathogen interaction | Organismal Systems | Environmental adaptation | 0.572 | 1.62E−20 |
| ko00940 | Phenylpropanoid biosynthesis | Metabolism | Biosynthesis of other secondary metabolites | 0.561 | 1.29E−05 |
| ko04016 | MAPK signaling pathway - plant | Environmental Information Processing | Signal transduction | 0.522 | 5.41E−05 |
| ko00053 | Ascorbate and aldarate metabolism | Metabolism | Carbohydrate metabolism | 0.574 | 0.003 |
| ko00904 | Diterpenoid biosynthesis | Metabolism | Metabolism of terpenoids and polyketides | 0.653 | 0.003 |
| ko00965 | Betalain biosynthesis | Metabolism | Biosynthesis of other secondary metabolites | 0.694 | 0.005 |
| ko00591 | Linoleic acid metabolism | Metabolism | Lipid metabolism | 0.604 | 0.007 |
| ko04712 | Circadian rhythm- plant | Organismal Systems | Environmental adaptation | 0.532 | 0.007 |
| ko00944 | Flavone and flavonol biosynthesis | Metabolism | Biosynthesis of other secondary metabolites | 0.743 | 0.011 |
| ko00052 | Galactose metabolism | Metabolism | Carbohydrate metabolism | 0.529 | 0.027 |
| ko00561 | Glycerolipid metabolism | Metabolism | Lipid metabolism | 0.533 | 0.033 |
| ko00941 | Flavonoid biosynthesis | Metabolism | Biosynthesis of other secondary metabolites | 0.557 | 0.033 |
| ko00073 | Cutin, suberine and wax biosynthesis | Metabolism | Lipid metabolism | 0.582 | 0.037 |
| ko00903 | Limonene and pinene degradation | Metabolism | Metabolism of terpenoids and polyketides | 0.649 | 0.037 |
| ko00900 | Terpenoid backbone biosynthesis | Metabolism | Metabolism of terpenoids and polyketides | 0.545 | 0.045 |
| ko00945 | Stilbenoid, diarylheptanoid and gingerol biosynthesis | Metabolism | Biosynthesis of other secondary metabolites | 0.604 | 0.045 |

**Notes.**
[a]Large categories only includes cellular processes, environmental information processing, genetic information processing, metabolism and organismal systems.
[b]Subcategories under each large category.

between the expression levels of the selected genes in the RT-qPCR analysis and those of the transcriptome analysis, and we inferred that it might be due to the different growth stages of the experiment's materials. Photosynthesis is an important metabolic process in plants, and its strength has an important effect on plant growth, development, and stress resistance. *Li & Zhang (2000)* found that the lower leaves of the fast-growing triploid *P. tomentosa* clones could maintain a higher photosynthetic rate when measuring the leaf net photosynthetic rate of diploid and triploid *P. tomentosa*. Photosynthesis includes a series of complex reactions in which carbon fixation is a central link in the regulation of photosynthesis (*Feng et al., 2006*). We compared the data of carbon fixation pathways in previous photosynthetic organism kiwifruit (*Li et al., 2019*) and selected the up-regulated malate dehydrogenase (*MDH*) for the RT-qPCR validation, and the expression levels of *MDH* gene were up-regulated in triploid *P. tomentosa* plants, compared to the diploid ones. With the possible exception of the stomatal dimension, the response to polyploidy can be very variable and complex. It has been proved in oilseed rape (*Bancroft et al., 2011*), sugarcane (*Manners & Casu, 2011*), cotton (*Rambani, Page & Udall, 2014*), wheat

**Table 5** Information of growth-related differential genes.

| KEGG pathway | | GIDs | Gene annotation | Gene function |
|---|---|---|---|---|
| Ko04075<br>Plant hormone<br>signal transduction | AUX1 | XM_011049129.1 | *Populus euphratica* auxin transporter-like protein 2 (LOC105141777), transcript variant X1, mRNA | Cell enlargement, plant growth |
| | GH3 | XM_011032075.1 | *P. euphratica* probable indole-3-acetic acid-amido synthetase GH3.1 (LOC105129841), mRNA | Indirect regulation of cell expansion and plant growth |
| | A-ARR | XM_011002137.1 | *P. euphratica* two-component response regulator *ARR8* (LOC105108010), mRNA | Cell division and bud formation |
| | CYCD3 | XM_011010474.1 | *P. euphratica* cyclin-D3-3-like (LOC105114061), mRNA | Indirect regulation of cell dvision and cell elongation |
| | ABF | XM_011043006.1 | *P. euphratica* ABSCISIC ACID-INSENSITIVE 5-like protein 2 (LOC105137304), transcript variant X2, mRNA | Participation in stress resistance processes such as low temperature, high salt and oxidation stress, and involvement in plant growth and development |
| Ko00710<br>Carbon fixation in<br>photosynthetic organisms | MDH | XM_011004743.1 | *P. euphratica* oligopeptide transporter 4 (LOC105109875), mRNA | Participation in the TCA cycle, photosynthesis, C4 cycle, and other metabolic pathways to promote plant growth and development |

(*Leach et al., 2014*), kiwifruit (*Li et al., 2019*), rice (*Shenton et al., 2020*) and numerous studies in Arabidopsis species, some of which specifically consider triploidy (*Fort et al., 2016*; *Fort et al., 2017*; *Pacey, Maherali & Husband, 2019*). Related to the phenomenon, the gene expression levels were not upregulated in any polyploid plant, and over-, under- or mixed-expression of genes were found in the polyploid plant (*Osborn et al., 2003*; *Gutierrez-Gonzalez & Garvin, 2017*; *Li et al., 2019*). Some genes in polyploid plants were upregulated while others were downregulated compared to diploid plants, which was also found in this study. It illustrated that the allopolyploidy –in particular in *P. tomentosa* –also alter the gene expression profile and levels as well as those in autopolyploid plants, compared to their diploid relatives. In this study, the expression levels that related to the growth genes such as *MDH* and *CYCD3* in triploid *P. tomentosa* were higher than those of diploid *P. tomentosa*.

*MDH* is mainly involved in the metabolism of plant photosynthesis (*Sawada et al., 2002*). The main function of the protein encoded by the *MDH* gene is to control the carbon dioxide levels during photosynthesis (*Sawada et al., 2002*). Higher expression levels of the *MDH* gene was proved to be related to higher rates of photosynthesis (*Kandoi, Mohanty & Tripathy, 2018*), which in turn contributes to higher timber yield. CYCLIN D3 (*CYCD3*) is a cell-cycle gene, and overexpression of CYCLIN D3;1 (CYCD3;1) in transgenic plants can increase mitotic cycles and reduce endocycles (*Menges et al., 2006*). *CYCD3* was found to regulate cambial cell proliferation and secondary growth, and the protein encoded by the *CYCD3* gene is required for normal vascular development in Arabidopsis (*Collins, Maruthi & Jahn, 2015*). Previous studies have indicated a relationship between faster

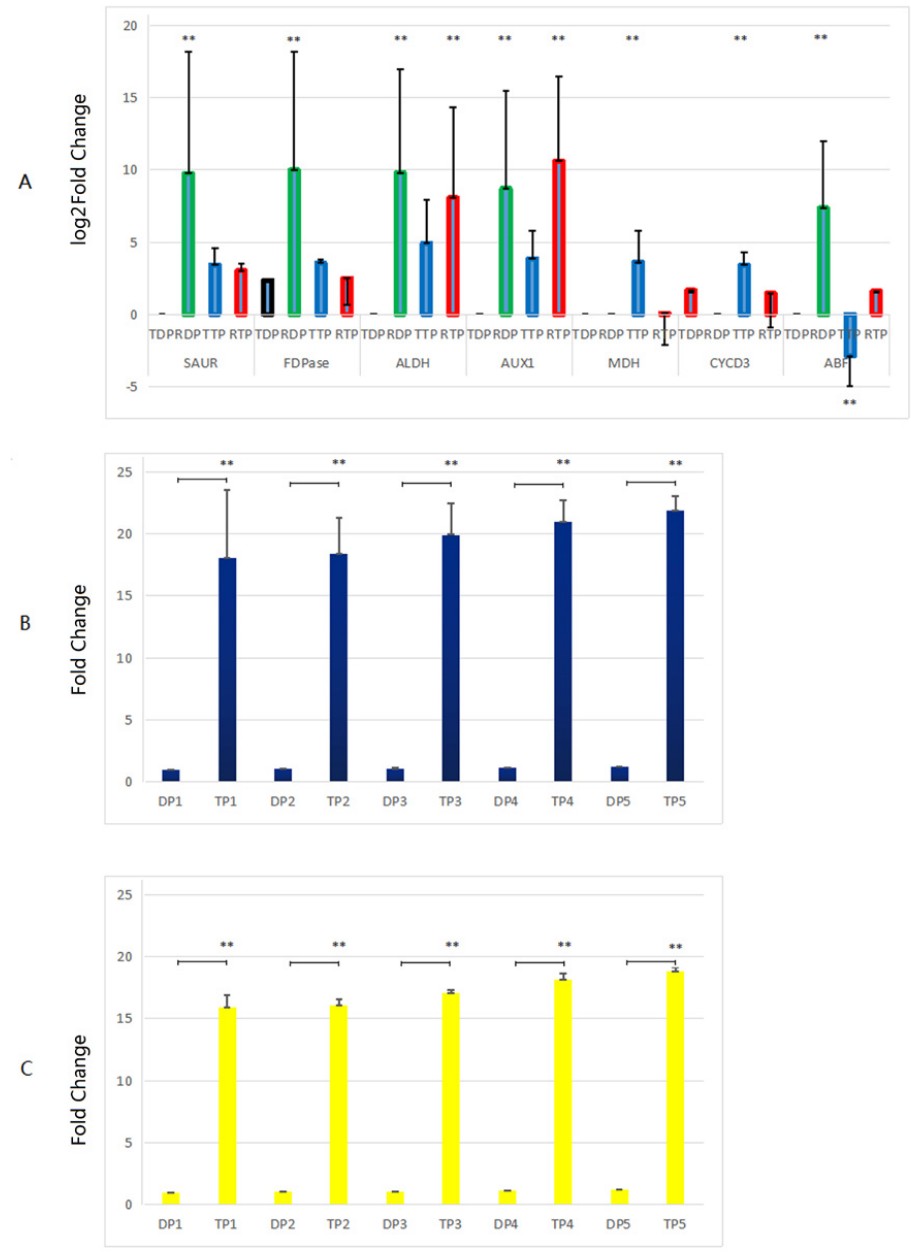

**Figure 5   Comparison of the transcriptome analysis and the RT-qPCR analysis.** (A) TDP, transcriptome data of diploid plants; RDP, RT-qPCR data of diploid plants; RTP, RT-qPCR data of triploid plants; TTP, transcriptome data of triploid plants. (B, C) DP, Diploid plants; TP, triploid plants. These plants were tissue culture plants and planted in greenhouse. 1, 1 month old; 2, 4 months old; 3, 7 months old; 4, 10 months old; 5, 13 months old. **, the difference is very significant (*P*-value < 0.01). RT-qPCR was performed on 3 diploid and 3 triploid plants, which were driven from the same tree used for transcriptome analysis, normalized with housekeeping gene *EF1α*, repeated 3 times. The $2^{(-\Delta\Delta Ct)}$ method was utilized to process the data (*Liu et al., 2018*). Because the difference of the values in (A) was too large, the *y*-axis of (A) was changed to a log scale.

growth and the increased expression of the *CYCD3 gene*. Hence, the upregulation of *MDH* and *CYCD3* in the triploids is biologically meaningful.

There were many DEGs between the diploid and triploid poplar plants that were enriched for a plant-pathogen interaction pathway, stress resistance and several growth-related transcripts too. Genes in the plant-pathogen interaction pathway are known to be diverse and large in plants and are known to be involved in reproduction isolation across Arabidopsis and Poplars too (*Liao et al., 2014*; *Qian et al., 2018*). In this study, one differential gene of the developmental process involved in reproduction was also found in the plant-pathogen interaction pathway. Further, the enrichment of differential expression transcripts in this category could provide useful information to tree breeders who intend to generate heterotic F1s. The pathway enrichment approach also shows several other interesting candidates that would be worth elaborating on in the future relating to studies demonstrating higher stress resistance in the triploid poplar.

## CONCLUSION

A total of 32,661 DEGs were identified in triploid and diploid Chinese white poplar, of which 15,690 were up-regulated and 16,971 were down-regulated in triploidy compared to diploidy. Through the comprehensive analysis of GO functional enrichment analysis and the pathway functional annotation of transcriptome data of diploid and triploid *P. tomentosa*, no significantly enriched entries and pathways related to growth were found. Compared to diploidy, the growth-related genes were found to be up-regulated and down-regulated in the natural diploid and triploid *P. tomentosa* trees. Although the expression levels of genes were unstable in the different environments and different growth stages, the expression levels of *MDH* and *CYCD3* in triploid *P. tomentosa* were higher than those of diploid *P. tomentosa* in young tree tissue, which was consistent with the values calculated using the transcriptome data.

## ACKNOWLEDGEMENTS

The manuscript was proofread by Proofed Inc. (UK).

### Funding

The project was supported by the National key R & D Plan for the 13th Five-Year Plan Project of China (Grant No. 2016YFD0600102) and the National Natural Science Foundation of China (Grant No. 31760450). The funders had no role in study design, data collection and analysis, decision to publish, or preparation of the manuscript.

### Grant Disclosures

The following grant information was disclosed by the authors:
The National key R & D Plan for the 13th Five-Year Plan Project of China: 2016YFD0600102.
The National Natural Science Foundation of China: 31760450.

## Competing Interests

The authors declare there are no competing interests.

## Author Contributions

- Wen Bian performed the experiments, prepared figures and/or tables, and approved the final draft.
- Xiaozhen Liu performed the experiments, analyzed the data, prepared figures and/or tables, authored or reviewed drafts of the paper, and approved the final draft.
- Zhiming Zhang analyzed the data, authored or reviewed drafts of the paper, and approved the final draft.
- Hanyao Zhang conceived and designed the experiments, performed the experiments, authored or reviewed drafts of the paper, and approved the final draft.

## Data Availability

The RNA-Seq data are available at the Genome Sequence Archive of the Beijing Institute of Genomics (BIG) Data Center: PRJCA002269.

## Supplemental Information

Supplemental information for this article can be found online at http://dx.doi.org/10.7717/peerj.10204#supplemental-information.

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
