# Peer review of "Transcriptome analysis of diploid and triploid Populus tomentosa"

_PeerJ, doi:10.7717/peerj.10204_

## Round 0.1 · original submission · Major Revisions

Both reviewers made several important points that need to be addressed. Both highlighted the need to expand on details in the methods section. I also agree with reviewer 2 that additional effort expanding the background/introduction would be helpful, and encourage you to address reviewer 1's notes about the experimental design./results. I do not, however, feel that additional experiments (like qRT-PCR) are necessary.

Reviewer 1 ·

Basic reporting

Bian et al. use gene expression dataset obtained from RNAseq and qRT-PCR to investigate patterns of gene expression difference between diploid and triploid species of P. tomentosa. Their study is based on previous work showing differences in growth and stress resistance between the two species of Populus. They noted several transcripts that were DE between the diploid and triploid Poplar that were enriched for plant-pathogen interaction pathway, stress resistance and several growth related transcripts too.

Overall, this manuscript describes patterns of gene expression in the diploid and triploid species of poplar. However, I think because the authors were very focussed on finding growth related transcripts being upregulated in the triploid, they have missed the opportunity to explore certain pathways that may not directly be related to growth and superior performance but can have key implications for tree breeders. Besides, superior growth in the triploid poplar need not always be related to upregulation of growth related transcripts across all developmental stages. With regard to exploring other pathways, their study shows the plant-pathogen interaction pathway to be enriched among sets of transcripts that were DE. Genes in this pathway are known to be diverse and large in plants and are known to be involved in reproduction isolation across Arabidopsis and Poplars too. While investigating neo-functionalisation and sub-functionalisation of genes belonging to this pathway in the triploid Poplar maybe beyond the scope of this work, it would be useful to speculate on this in the discussion. A simple summary of how many genes in this pathway have zero expression in the triploid poplar or are associated with different GO node terms relative to the diploid would be good. Further, the enrichment of DE transcripts in this category could provide useful information to tree breeders intending to generate heterotic F1s. The pathway enrichment approach also shows several other interesting candidates that would be worth future elaborating on and relating to previous studies demonstrating higher stress resistance noted in the triploid poplar. Another place to elaborate would be the DE of MDH & CYCD3 genes as evaluated using qRT-PCR. I think it would be useful to relate their finding of higher expression levels to higher rates of photosynthesis which in turn contributes to higher timber yield. Maybe relate to other studies that have demonstrated a relationship between the rate of photosynthesis and faster growth and higher timber yield or have evaluated the rate of growth in mutants of MDH and CYCD3.

I would also have liked to see a qRT-PCR on lignin production related transcripts since it is a key trait used by tree breeders. If that is not possible, evaluating patterns of DE for some of the lignin production related transcripts using the RNAseq data would also be useful.



Specific and minor comments:

Line 49: this sentence seems incomplete, should it be a continuation of the previous sentence and not a new sentence.
Line 58: please provide a reference for genome duplication due to hybridization.
Consider changing line 73 to read as : RNA was extracted using the Tiagen kit.
Line 74: change “performed” to “generated”
Please provide the settings used while running Trimmomatic and Soapnuke. For instance, was trimming of polyA tails conducted, what was Q score cutoff, length cutoff etc. in Trimmomatic.
Line 114: This is the first instance where the authors talk about CDS & SSR, please mention in the methods section how these were detected.
Were the GO annotations performed using protein sequences or using nucleotide? I am assuming protein sequences were used for this.
Line 124: Detection of transcription factor families is not mentioned in the methods. Is this based on the annotations, if so please clarify.
It is unclear to me what the authors mean by “involved in the transcriptome” mentioned at line 126.
Line 137: This sentence is unclear. Does 22k refer to the set of DE transcripts that had GO annotations, or does it refer to the set that was not DE? Please clarify.
Line 150-153: Not clear.
Line 159: This is confusing. Earlier you mention that there are in total 32k DE transcripts, so what does this 12k refer to?
Line 177-178: Please elaborate what you mean by “was obviously different from transcriptome analyses”.
Line 193: Please provide a citation for this statement.
Please rewrite line 193 to 196. In line 193 it is not clear whether the authors are referring to larger leaf sizes for polyploid plants. Similarly for line 194, consider rewriting to:
Huang et al (1990) have demonstrated larger branches, leaves and fruits in the triploid variety of pear compared to the diploid variety.
Line 202: what are the authors comparing to?
Line 214-217: These sentences as written are not clear. Please consider elaborating what you mean by cryptic. It is also not clear how the various studies listed here are related to gene expression differences noted in your study.
Table 2 & 3 needs to be formatted correctly. Also consider using superscripts to describe what level1 and level2 are.
Please check if fig 3 is representing Up and down regulated transcripts after FDR.
Fig 6: It is very difficult to see the bars for expression levels using the transcriptome dataset, which is expected given the higher sensitivity of qRT-PCR. I would suggest changing the y-axis to a log scale or presenting these as two separate panels. The y-axis is also not labelled.

Experimental design

Are D1, D2, D3 & T1, T2, T3 representing biological replicates or technical replicates? Please clarify at the first mention and also in the methods for DE.

There are also several other aspects of the methods that need to be elaborated on and I have listed them above under "specific & minor comments".

Validity of the findings

--FPKM not ideal for across sample comparisons (unless the cross sample scaling was performed prior to getting FPKM). FPKM is ok to compare gene expression within the same sample. I would suggest using TMM which is easy to implement and robust.
See:
https://www.ncbi.nlm.nih.gov/pmc/articles/PMC3608160/
https://www.ncbi.nlm.nih.gov/pmc/articles/PMC5411036/
https://genomebiology.biomedcentral.com/articles/10.1186/gb-2010-11-3-r25


--Please provide more details for the DE section of the methods. It is important to mention that the data was normalised and the approach used for normalisation and whether the transcripts were filtered for minimum count prior to assessing differential expression.

--I am curious as to why the authors decided to use phyper against several other methods specifically designed for conducting GO category enrichment. But more importantly, please mention whether the test conducted was two tailed or one tailed for enrichment. Based on the figure, it seems the test was one tailed.

Reviewer 2 ·

Basic reporting

Although the authors used a proofreading service, this manuscript could benefit from another proofreading by someone familiar with the field. The English used in the article is, in general, good. However, there are enough instances where the meaning of the authors is ambiguous that the overall clarity should be addressed.



The authors primarily focus on the need for better approaches to crop improvement of tree species from an applied point of view (i.e. discussing the superiority of triploid P. tomentosa for timber qualities and growth vigor). Given that PeerJ has a relatively broad audience, the authors may need to explain some assertions they currently take for granted. What defines good wood quality? Fast growth and long life as compared to what? Other trees in northern China? Trees worldwide? Commercial trees? The second paragraph of the introduction does a better job of describing the difficulties of performing genetic improvement of tree crop species, but I think it would benefit from a considerably more comprehensive literature search.

In general, there is not sufficient background provided in the introduction, and references are sparse. Additional scholarship should be done, particularly focused on diploid vs polyploid traits in crop species. The third paragraph describes differences between triploid and diploid P. tomentosa, but misses an opportunity to better motivate their work. There is a wide literature comparing diploid and polyploid individuals for a variety of crop species that would be relevant here, but the authors choose to focus on P. tomentosa alone. A broader discussion of why polyploid plants often exhibit traits that are commercially beneficial, especially with respect to differences in gene expression, would be beneficial here.

Minor comments/suggestions:
Line 35: Briefly explain what good wood quality is.
Line 45: "the tree transcriptome" or "tree transcriptomes"
Line 51: omit ploidy
Line 52 and line 182: although not exactly wrong, "obviously" is not a commonly-used word in most scientific literature (generally one's data should make something obvious). I would omit this.
Line 57-58: "the transcriptome analysis was subjected to an analysis of the differences between diploid and triploid P. tomentosa" is a bit awkward and unclear. Differences in what?
Line 68: "o'clock"
Line 195: "found to be huge in triploids on branches" this is unclear
Line 197: "allotriploidy of P. tomentosa was better than diploidy" It would be better to say that the triploid had greated values of these traits. "better" implies a value judgment.

It is definitely possible that I did something incorrectly, because I'm not very familiar with the BIG database, but I could not download the raw data. However, I can see that this project is listed in the database. Someone with greater familiarity may be able to pull down the raw reads.

Experimental design

The methods as written are rather sparse, and it is difficult to get a good idea of what was done, or to understand if the protocols used are appropriate. As currently written, they would not be sufficient to replicate the experiment.

Where the trees native to the area sampled, or where they planted? If planted, is there any way to determine where these genetic stocks originally come from? Are the trees precisely one-year-old, or is there variability in their ages (and if so, how much?). How much material was sampled from each individual? The methods also appear in some cases to be out of order. Was RNA extraction done by the authors? What were the operating parameters used for data filtering and analysis? How was annotation performed? What pipeline was used (and if none was used, how was annotation done)? How was tissue culture performed on plants to be used for RT-qPCR? Although the authors do refer to other papers for some methods, a brief description of the methods (RNA extraction, RT-qPCR protocol, data processing)

Line 69: "top-to-bottom fifth leaf". This phrase is somewhat confusing, and if I am interpreting it correctly, raises a question about the protocol. Does this mean the fifth leaf down from the tip of a branch, or the fifth leaf up the stem from the base? If it means the fifth leaf down from the top, are the authors certain they sampled analogous leaves from each individual? Generally, similar experiments count leaves or nodes from the base of the plant, but it's not entirely clear what the authors did and if it was appropriate.

Line 210-213: "We compared the data of carbon fixation pathways in previous photosynthetic organisms and selected the up-regulated malate dehydrogenase (MDH) for the RT-qPCR validation, and the expression levels of MDH gene were up-regulated in triploid P. tomentosa plants, compared to the diploid ones." Which photosynthetic organisms?

Validity of the findings

The discussion and conclusions, as with the rest of the paper, are quite short, and do not spend enough time discussing the results of the experiments and how they fit into the broader literature. For example, it seems unusual that P. tomentosa would match the assembled transcriptome to such a low degree (3.26%) given that the individuals studied are supposedly P. tomentosa. The authors do not mention this in their discussion, but I feel it merits pointing out and explaining.

Although the authors do spend some time mentioning previous work on polyploids, they mainly list species on which similar studies have been done. A more thorough description of which studies found over-, under- or mixed expression would be nice, as would some discussion and speculation on the role of allopolyploidy in particular in P. tomentosa. Some discussion on if the authors think upregulation of MDH and CYCD3 in the triploids is biologically meaningful, or simply due to chance would be useful in the discussion.

Additional comments

This work is interesting, and will be useful to tree geneticists and tree crop specialists, and shows how variable gene expression can be in young tree tissues. However, the brevity of the manuscript means that a lot of useful detail and important scholarship is missing. I encourage the authors to add additional detail to the methods section in particular.

---

## Round 0.2 · Minor Revisions

Thank you for the revisions. The manuscript is indeed improved. I would encourage you to read through carefully (or hire a service to do so) for grammar -- in several places the grammar still leads to ambiguous sentences that are difficult to understand.

I'm not sure why you can't answer the reviewer question about genes with zero expression in the triploid? Surely some of the DEGs must have zero expression in triploid? I also may have missed it but don't see the reviewer's request to include different GO terms between the diploid and triploid?

Please be sure your plots include information on what the whiskers represent (standard errors? deviation? confidence interval?).

---

## Round 0.3 · accepted · Accept

Thank you for your attention to these revisions!